# Bleaching of *Idesia polycarpa* Maxim. Oil Using a Metal-Organic Framework-Based Adsorbent: Kinetics and Adsorption Isotherms

**DOI:** 10.3390/foods14050787

**Published:** 2025-02-25

**Authors:** Yiyang Dong, Chengming Wang, Yu Gao, Jing Xu, Hongzheng Ping, Fangrong Liu, Aifeng Niu

**Affiliations:** 1College of Food Science and Technology, Huazhong Agricultural University, Wuhan 430070, China; dongyiyang@webmail.hzau.cn (Y.D.); gaoyu@webmail.hzau.cn (Y.G.); xujing0621@webmail.hzau.cn (J.X.); phz@webmail.hzau.cn (H.P.); l15884925210@webmail.hzau.cn (F.L.); niuaifeng@webmail.hzau.cn (A.N.); 2Key Laboratory of Environment Correlative Dietology, Huazhong Agricultural University, Ministry of Education, Wuhan 430070, China

**Keywords:** *Idesia polycarpa* Maxim. oil, bleaching, MOF, MIL-88B(Fe), adsorption kinetics and isotherms

## Abstract

*Idesia polycarpa* Maxim. is a woody oil crop with great potential for edible oil production. While crude oil is rich in pigments, traditional bleaching methods have limited effectiveness in improving its color. In this study, a metal-organic framework (MOF) material, MIL-88B(Fe), was synthesized and used for the bleaching of *Idesia polycarpa* Maxim. oil. The adsorption selectivity of MIL-88B(Fe) and the adsorption process of carotenoids and chlorophyll were investigated. The results demonstrated that the synthesized MIL-88B(Fe) exhibited excellent bleaching capability, achieving a bleaching rate of 97.67% in 65 min. It showed a strong adsorption effect on pigments, particularly carotenoids. The content of lutein decreased from 118.27 mg/kg to 0.01 mg/kg after 65 min of bleaching. The squalene and phytosterol contents in the oil were minimally affected by the bleaching process, while the free fatty acid content slightly increased due to the high reaction temperature and the adsorbent properties. The adsorption process of MIL-88B(Fe) was best described by a pseudo-first-order kinetic model, indicating that the adsorption was a spontaneous and endothermic chemical process. Moreover, MIL-88B(Fe) demonstrated good safety and reusability, making it a promising novel adsorbent for the bleaching of *Idesia polycarpa* Maxim. oil and other oils with a high pigment content for the vegetable oil industry.

## 1. Introduction

*Idesia polycarpa* Maxim. is an evergreen tree belonging to the monotypic genus of the family Clusiaceae, native to southern China and Japan [1]. The fruit is the primary economic source of *Idesia polycarpa* Maxim, which is characterized by high oil content. Additionally, the fruit contains various bioactive compounds, such as tocopherols, squalene, sterols, peptides, and polyphenols [2], exhibiting notable antioxidant and anti-inflammatory activities [3,4]. In the meanwhile, it contains a variety of undesirable substances such as pigments, which may bring consumers unpleasant colors and odors. For example, carotenoids are natural and fat-soluble pigments, which give *Idesia polycarpa* Maxim. oil the vibrant reddish-brown color. To meet the needs of industrial production, these impurities must be removed at various steps in the refining processes, including degumming, neutralization, washing, drying, bleaching, filtration, and deodorization.

Bleaching represents a critical step in the refining process of vegetable oils, mainly by removing pigments, such as carotenoids and chlorophyll, to improve the oil’s appearance [5]. Chlorophyll tends to decompose into pheophytin under heat, leading to opaque and dark oil [6]. The bleaching process usually employs activated adsorbent materials, such as bleaching earth, activated carbon, and silica, to remove pigments and other impurities from the oil [7,8]. However, due to the deep red color of *Idesia polycarpa* Maxim. oil, traditional bleaching methods often require large amounts of adsorbents, which may result in partial oil loss along with the adsorbents, low oil recovery rates, and suboptimal bleaching performance. Therefore, developing new and efficient materials has become an immense need in the bleaching of *Idesia polycarpa* Maxim. oil.

Metal-organic frameworks (MOFs) have emerged as an extensive class of organic-inorganic hybrid porous materials characterized by tunable pore structures, large surface areas, adjustable organic ligands, and abundant unsaturated sites [9]. The cage-like structures, high surface areas, and pore volumes make them widely applicable in gas adsorption, storage, separation, catalytic reactions, drug delivery, and chemical sensing [10,11]. In recent years, the high adsorption capabilities of MOFs have attracted significant attention in the refining of vegetable oils [12]. Studies have shown that aluminum, zinc, and titanium MOFs based on terephthalic acid can effectively adsorb free fatty acids and peroxides in unrefined vegetable oils, thereby improving their physicochemical properties [13]. For example, Du et al. used the synthesized MOF-235 to simultaneously remove aflatoxins and zearalenone from vegetable oils, demonstrating the significant efficacy in removing target residues, along with safety and reusability [14]. Yılmaz and Erden tested seven MOFs as adsorbents for purifying crude degummed sunflower oil and compared their effects with three natural clays [15]. It was found that these MOFs could improve the physicochemical properties of the oil, suggesting the potential of MOFs as adsorbents for crude oil purification. They also showed that MOF treatment could efficiently improve the color value of crude oil compared to natural bleaching earths such as C.B.E. and sepiolite. Similarly, Jia et al. developed MOF-aerogel composites for oil decolorization, with MOF-aerogel-2 showing high efficacy in carotenoid removal, which was comparable to a commercial activated bleaching earth [16]. However, due to the high carotenoid content in *Idesia polycarpa* Maxim. oil, commercial activated bleaching earth has shown unsatisfactory bleaching effects. Therefore, an efficient MOF material capable of adsorbing pigments in *Idesia polycarpa* Maxim. oil is required.

Iron-based MOFs (Fe-MOFs) have gained attention as environmentally friendly alternatives owing to their ready availability, low cost, non-toxic metal sources, high porosity, framework, and flexibility properties [17,18]. The MIL series serves as one of the earliest Fe-MOFs and features highly controllable pore structures, thermal stability, and chemical stability [19]. Among various Fe-MOFs, MIL-88B(Fe) consists of octahedral structures formed by iron ions coordinated with terephthalic acid ligands, exhibiting a “breathing” effect that allows the pore size to adjust in response to external stimuli. This property provides MIL-88B(Fe) with a large channel structure and high tunability [20]. As an adsorbent, it presents extraordinary adsorption capacity toward various pollutants and dyes, even in a scaling-up condition [21,22]. Moreover, MIL-88B(Fe) as one of the Fe-based MOFs offers the characteristics of chemical stability, easy synthesis, low toxicity, and environmentally friendly [23].

In this study, we synthesized MIL-88B(Fe) with a porous structure and good stability for the bleaching of *Idesia polycarpa* Maxim. oil for the first time. The adsorption selectivity of MIL-88B(Fe) for trace substances in the oil, the kinetics, isotherms, and thermodynamics of the adsorption process, the regeneration bleaching ability and reusability, as well as the effects of bleaching on the physicochemical properties, fatty acid composition, and Fe content in *Idesia polycarpa* Maxim. oil after bleaching with MIL-88B(Fe), were evaluated. These provide important theoretical support for the bleaching of *Idesia polycarpa* Maxim. oil using MIL-88B(Fe), further offering a new and efficient method for the oil refining industry.

## 2. Materials and Methods

### 2.1. Materials

The *Idesia polycarpa* Maxim. crude oil was purchased from Enshi, Hubei (China), and used in the bleaching procedure after laboratory degumming and deacidification. All chemicals used were of chromatographic grade. FeCl_3_·6H_2_O (98%), terephthalic acid (H_2_BDC, 98%), NaOH (98%), N,N-dimethylformamide (DMF, 98%), cyclohexane, potassium hydroxide, ascorbic acid, and dibutylhydroxytoluene were obtained from Shanghai McLean Biochemistry and Technology Co., Ltd. (Shanghai, China). Diethyl ether, n-hexane, anhydrous ethanol, methyl tert-butyl ether, anhydrous sodium sulfate, acetonitrile, methanol, 2-bromoacetophenone, acetic acid, triethanolamine, and acetone were purchased from Sinopharm Chemical Reagent Co., Ltd. (Shanghai, China). Standards including β-carotene (≥95.0%), lutein (≥98.0%), cholesterol (≥99.0%), and heptadecanedioic acid methyl ester (≥99.0%) were obtained from Shanghai Yuanye Biotechnology Co., Ltd. (Shanghai, China).

### 2.2. Synthesis and Characterization of MOF Adsorbents

MIL-88B(Fe) was synthesized following the method described in a previous study [24]. In brief, 1.62 g of iron (III) chloride hexahydrate (FeCl_3_·6H_2_O) and 1.67 g of terephthalic acid (H_2_BDC) were dissolved in 30 mL of N,N-dimethylformamide (DMF). Subsequently, 4 mL of 2 M NaOH solution was slowly added to the mixture under stirring. The resulting solution was transferred to a 100 mL Teflon-lined autoclave and heated at 100 °C for 12 h. After cooling naturally to room temperature, the product was centrifuged. The orange product was washed three times with DMF, methanol, and deionized water, respectively, and then dried in a vacuum oven at 110 °C for 12 h to obtain the MIL-88B(Fe) material.

The crystal structure and crystallinity of the synthesized MIL-88B(Fe) were analyzed using X-ray diffraction (XRD, D8 Advance, Bruker, Karlsruhe, Germany). The morphology and size of MIL-88B(Fe) were observed using a scanning electron microscope (SEM, Gemini, Carl Zeiss, Oberkochen, Germany). The chemical functional groups and molecular structure of MIL-88B(Fe) were determined using a Fourier-transform infrared spectrometer (FT-IR, IS50, Opus, Suzhou, China). The thermal stability of the sample material was assessed by thermogravimetric analysis (TGA, TG 209F3, Netzsch, Scandinavia, Germany).

For XRD, phase analysis was conducted using a D8 Advance X-ray diffractometer under the following experimental conditions: tube voltage and current set at 40 kV and 40 mA, respectively; a scanning rate of 3°/min; a scanning angle (2θ) range from 5° to 30°; and a step size of 0.02°. The obtained experimental XRD data were compared with simulated data generated from crystal structures retrieved from the Crystallography Open Database (COD) or the Cambridge Crystallographic Data Centre (CCDC). For SEM, a small amount of the sample was directly adhered to conductive tape and coated with gold using a Quorum SC7620 sputter coater (Sussex, UK) for 45 s at a current of 10 mA. Subsequently, the sample morphology was characterized using a ZEISS GeminiSEM 300 scanning electron microscope. The imaging was performed at an accelerating voltage of 3 kV, and the SE2 secondary electron detector was utilized for capturing the morphological features. For FTIR, the chemical functional groups and molecular structures of the prepared materials were characterized using an IS50 Fourier-transform infrared (FTIR) spectrometer, produced by Suzhou Opus Plasma Technology Co., Ltd. (Suzhou, China). Potassium bromide (KBr) was used as the background, and the scanning wavelength range was set from 400 to 4000 cm^−1^. For TGA, the thermal stability of the sample material was evaluated using a TG 209F3 thermogravimetric analyzer (Netzsch, Germany). The test procedure involved heating the material from 25 °C to 800 °C in a flow of air at a heating rate of 5 °C/min while monitoring the weight loss process of the material.

### 2.3. Bleaching Experiments

The bleaching process was conducted by adding MIL-88B(Fe) material ranging from 5% to 9% (*w*/*v*, %) to a flat-bottom flask containing 10 g of *Idesia polycarpa* Maxim. oil [25]. The mixture was stirred in a water bath at temperatures ranging from 80 °C to 120 °C for 15 to 85 min. After the reaction, the oil sample was separated by vacuum filtration from the adsorbent to obtain the bleached *Idesia polycarpa* Maxim. oil, which was used for the next analysis. The unbleached oil was used as a control.

### 2.4. Determination of Carotenoids, Lutein, and β-Carotene

About 1 g of *Idesia polycarpa* Maxim. oil was dissolved in 5 mL of n-hexane, and the absorbance of the solution at 445 nm was measured using a UV-Vis spectrophotometer (UV-1800, Shimadzu, Kyoto, Japan) according to the reported method [26]. The total content of the carotenoids was calculated based on the standard curve using β-carotene as the standard.

The determination of lutein and β-carotene was performed using high-performance liquid chromatography (HPLC) (LC-20A, Shimadzu, Japan). The saponification process involved adding 1 g of ascorbic acid, 0.2 g of BHT, 30 mL of anhydrous ethanol, and 30 mL of KOH to the sample (2 g), and saponifying in a thermostatic shaker at 54 °C for 45 min. The saponified solution was extracted with a 60 mL mixture of cyclohexane/ether/n-hexane (1:2:2, *v*/*v*/*v*, containing 0.1% BHT). The organic phase was washed with water until neutral, dehydrated by filtration through anhydrous sodium sulfate, and concentrated under reduced pressure to near dryness. The extract was dissolved in 0.1% BHT ethanol solution and diluted to a final volume of 10 mL. After filtration through a 0.45 µm membrane, the sample was analyzed using HPLC. Identification and quantification were performed by comparing chromatograms with those of standard substances. The chlorophyll content was determined based on the absorbance of the solution at 630, 670, and 710 nm [27] and was calculated by below Equation (1):(1)$$Chlorophyllin\;content\left(mg/kg\right)=\frac{\left[A_{670}-\frac{\left(A_{630}+A_{710}\right)}{2}\right]V}{0.0964m}$$
where m is the mass of *Idesia polycarpa* Maxim. oil (g), V is the volume of diluted *Idesia polycarpa* Maxim. oil (mL).

### 2.5. Determination of Bleaching Efficiency

The deacidified *Idesia polycarpa* Maxim. oil was diluted 50 times with n-hexane. The optimal wavelength of *Idesia polycarpa* Maxim. oil was determined to be 445 nm using a UV spectrophotometer (UV-1800, Shimadzu, Japan). The absorbance of the oil was measured at 445 nm before and after bleaching using different adsorbents to calculate the bleaching efficiency (BE) (Equation (2)). The solvent n-hexane was used as a control.(2)$$BE\left(\%\right)=\frac{A_{0}-A_{1}}{A_{0}}\times 100\%$$
where $A_{0}$ is the absorbance of *Idesia polycarpa* Maxim. oil before bleaching; $A_{1}$ is the absorbance of *Idesia polycarpa* Maxim. oil after bleaching.

### 2.6. Determination of Phytosterols and Squalene Content

The content of phytosterols and squalene in *Idesia polycarpa* Maxim. oil was determined according to EN ISO 12228-1: 2014 (Determination of individual and total sterols contents—Gas chromatographic method Part 1: Animal and vegetable fats and oils) using an HP-5 column (30.0 m × 250 μm × 0.25 μm) (Agilent, Colorado Springs, CO, USA). The cholesterol was used as the internal standard for quantification. The retention times of sterols and squalene were compared with those of reference standards for qualitative analysis. The injection port temperature was set at 300 °C, with high-purity nitrogen as the carrier gas and a split ratio of 15:1. The flow rate was 1.0 mL/min. The column temperature was maintained at 200 °C for 3 min, then increased at a rate of 2 °C/min to 220 °C, followed by a ramp of 20 °C/min to 300 °C, then it was held for 15 min. The detector temperature was set at 300 °C, and the injection volume was 1 μL.

### 2.7. Determination of Free Fatty Acid (FFA) Content

The FFA content in *Idesia polycarpa* Maxim. oil was determined using a Thermo U3000 UPLC system (Thermo Scientific, Lenexa, USA). A solution of 0.1 mL of 0.01 mol/L heptadecanoic acid in n-hexane was added to 100 mg of oil sample as the internal standard. The analysis was performed using an Agilent SB-Aq column (5 µm, 4.6 mm × 250 mm). The injection volume was 20 µL, the column temperature was set at 25 °C, and the flow rate was 1 mL/min. Detection was carried out at a wavelength of 246 nm with isocratic elution using methanol: acetonitrile: water (*v*/*v*/*v*, 80:10:10) for 13 min.

### 2.8. Kinetic Experiments

The kinetic experiments were conducted in a water bath at 100 °C, with the addition of 5% MIL-88B(Fe). The effect of bleaching time (0 to 85 min) on the adsorption of pigments by MIL-88B(Fe) was investigated. After bleaching, the content of carotenoids and chlorophyll in *Idesia polycarpa* Maxim. oil was determined using a UV-Vis spectrophotometer to identify the equilibrium time and equilibrium adsorption capacity. The content of carotenoids and chlorophyll adsorbed at different time intervals and equilibrium was calculated using Equations (3) and (4):(3)$$q_{t}=\frac{m_{0}\left(C_{0}-C_{t}\right)}{m_{a}}$$(4)$$q_{e}=\frac{m_{0}\left(C_{0}-C_{e}\right)}{m_{a}}$$
where:

$m_{0}$ is the mass of the deacidified *Idesia polycarpa* Maxim. oil (kg),

$m_{a}$ is the mass of MIL-88B(Fe) added (kg),

$C_{0}$ is the initial pigment content in the oil before bleaching (mg kg^−1^),

$C_{t}$ and $C_{e}$ are the pigment contents in the oil at time t and equilibrium (mg kg^−1^), respectively,

t is the bleaching time (min).

The adsorption kinetics of MIL-88B(Fe) for carotenoids and chlorophyll were studied using pseudo-first-order, pseudo-second-order, and intra-particle diffusion models [28], represented by Equations (5)–(7):(5)$$q_{t}=q_{e}\left(1-e^{-k_{1}t}\right)$$(6)$$q_{t}=\frac{q_{e}^{2}k_{2}t}{1+q_{e}k_{2}t}$$(7)$$q_{t}=k_{p}t^{0.5}+C$$
where:

$q_{t}$ and $q_{e}$ are the amounts of pigment adsorbed by MIL-88B(Fe) at time t and at equilibrium (mg g^−1^), respectively,

$k_{1}$ is the pseudo-first-order adsorption rate constant (min^−1^),

$k_{2}$ is the pseudo-second-order adsorption rate constant (g mg^−1^ min^−1^),

$k_{i}$ is the rate constant of the intra-particle diffusion model (mg g^−1^ min^−0.5^),

t is the bleaching time (min),

$C_{i}$ is the intercept of the intra-particle diffusion model.

### 2.9. Adsorption Equilibrium Experiments

To study the effect of MIL-88B(Fe) dosage on adsorption efficiency, 5%, 6%, 7%, 8%, and 9% (*w*/*v*) of MIL-88B(Fe) were added to 10 g of *Idesia polycarpa* Maxim. oil. The mixture was stirred at 100 °C for 35 min using a magnetic stirrer. After centrifugation, the bleached oil was analyzed for carotenoid and chlorophyll content. The adsorption isotherms were studied using the nonlinear forms of the Langmuir, Freundlich, Tempkin, and Toth models [29]:(8)$$Langmuir\;model\;:\;q_{e}=\frac{K_{L}q_{m}C_{e}}{\left(1+k_{L}C_{e}\right)}$$(9)Freundlich model : qe=KFCe1n(10)$$Tempkin\;model\;:\;q_{e}=\frac{RT}{b_{T}}\ln\left(A_{T}C_{e}\right)$$(11)$$Toth\;:\;q_{e}=\frac{K_{T}C_{e}}{{\left(a_{T}+C_{e}\right)}^{\frac{1}{t}}}$$
where:

$q_{e}$ is the amount of pigment adsorbed by the adsorbent at equilibrium (mg g^−1^),

$q_{max}$ is the maximum adsorption capacity (mg g^−1^),

$C_{e}$ is the pigment concentration in the oil at equilibrium (mg kg^−1^),

$K_{L}$ is the Langmuir constant (kg mg^−1^),

$K_{F}$ is the Freundlich constant (mg g^−1^) (kg mg^−1^)^1/n^,

n is a constant related to adsorption intensity,

T is the absolute temperature (K),

R is the universal gas constant (8.314 J mol^−1^ K^−1^),

$K_{T}$ is the Temkin constant corresponding to the maximum binding energy (kg mg^−1^),

$B_{T}$ is the Temkin constant related to adsorption heat (J mol^−1^),

$q_{m}$ is the maximum adsorption capacity in the Toth model (mg g^−1^),

$K_{T}$ is the Toth isotherm constant (kg mg^−1^),

t is a parameter that indicates the heterogeneity of the adsorbent (in 1/n).

### 2.10. Thermodynamic Experiments and Determination of Parameters

To examine the effect of temperature on the adsorption behavior of MIL-88B(Fe), 5% (*w*/*v*) MIL-88B(Fe) was added to 10 g of *Idesia polycarpa* Maxim. oil and stirred using a magnetic stirrer for 35 min at different temperatures (80, 90, 100, 110, and 120 °C). To determine whether the adsorption process is spontaneous, thermodynamic parameters were estimated from the best-fit parameters of the adsorption isotherms. The adsorption of pigments in *Idesia polycarpa* Maxim. oil was evaluated by the changes in Gibbs free energy (ΔG), enthalpy (ΔH), and entropy (ΔS). The thermodynamic parameters were calculated using the following equations [30,31]:(12)$$\ln\Delta k_{d}=\frac{\Delta S^{0}}{R}-\frac{\Delta H^{0}}{RT}$$(13)$$k_{d}=\frac{q_{e}}{C_{e}}$$(14)$$\frac{d\ln k_{d}}{dt}=\frac{\Delta H^{0}}{RT^{2}}$$(15)$$\Delta G^{0}=\Delta H^{0}-T\Delta S^{0}$$(16)$$\Delta G^{0}=-RT\ln k_{d}$$
where:

$K_{eq}$ is the thermodynamic equilibrium constant,

R is the universal gas constant (8.314 J mol^−1^ K^−1^),

T is the absolute temperature (K).

### 2.11. Determination of Reusability of MIL-88B(Fe)

To evaluate the reusability and stability of the prepared MIL-88B(Fe) for the bleaching of *Idesia polycarpa* Maxim. oil, the MIL-88B(Fe) used in the bleaching experiments was collected directly after adsorption and desorbed using Soxhlet extraction with anhydrous ethanol for 4 h. The desorbed MIL-88B(Fe) was then dried in a vacuum oven at 60 °C for 12 h before being used in the next cycle of experiments. Due to the presence of residual *Idesia polycarpa* Maxim. oil on the surface of the adsorbed MIL-88B(Fe), which was difficult to separate, Soxhlet extraction was found to be inefficient and time-consuming. Therefore, an alternative method was employed in subsequent cycles, where the adsorbed MIL-88B(Fe) was directly stirred in anhydrous ethanol at 60 °C for 1 h to achieve desorption, followed by drying before the next cycle of experiments. The bleaching efficiency (BE_n_) was calculated using the following equation (17):(17)$${BE}_{n}\left(\%\right)=\frac{A_{0}-A_{n}}{A_{0}}\times 100\%$$
where:

A_0_ is the absorbance of *Idesia polycarpa* Maxim. oil before bleaching, and A_n_ is the absorbance of the oil after bleaching with MIL-88B(Fe), which has been used n times.

### 2.12. Determination of Fe Content in Bleached Oil

The total Fe iron content in the bleached *Idesia polycarpa* Maxim. oil was quantitatively analyzed using Inductively Coupled Plasma Mass Spectrometry (ICP-MS) (7500 CX, Agilent, USA) [32]. The ICP-MS was operated under the following conditions: RF power of 1550 W, sample rinse time of 20 s, auxiliary gas flow rate of 1.00 L/min, nebulizer gas flow rate of 1.00 L/min, and peristaltic pump speed of 20 r/min. The Fe content in the sample was calculated using Equations (18) and (19):(18)$$C_{x}=\frac{C_{0}\times f\times V_{0}\times 10^{-3}}{m\times 10^{-3}}=\frac{C_{1}\times V_{0}\times 10^{-3}}{m\times 10^{-3}}$$(19)$$W=\frac{C_{x}}{10^{9}}\times 100\%$$
where:

m is the mass of the sample analyzed (g),

$V_{0}$ is the volume of the digested sample after dilution (mL),

f is the dilution factor,

$C_{0}$ is the concentration of the element in the test solution (µg/L),

$C_{1}$ is the concentration of the element in the original digestion solution (µg/L),

$C_{x}$ is the final concentration of the measured element (µg/kg),

W is the final content of the measured element expressed as a percentage (%).

### 2.13. Statistical Analysis

All experiments were conducted in triplicate, and the results are presented as mean ± standard deviation. Statistical analysis was performed using SPSS Statistics 26 (IBM, Armonk, NY, USA), and the experimental results were evaluated using Duncan’s multiple range test. Data visualization was carried out using Origin 2021 (OriginLab, Northampton, MA, USA).

## 3. Results and Discussion

### 3.1. Synthesis and Characterization of MIL-88B(Fe)

Figure 1a shows the XRD pattern of the synthesized MIL-88B(Fe), with high-intensity peaks identified within the 5° to 30° range. It can be observed that the XRD pattern of MIL-88B(Fe) matches well with the standard card (Simulated), indicating good consistency between the experimental data and simulated data in terms of diffraction peak positions. The XRD confirms the crystal structure of MIL-88B(Fe) material [33], with strong peaks demonstrating the crystalline nature of MIL-88B(Fe) [34]. Some diffraction peaks in the figure are shifted compared to the simulated data, possibly due to swelling of the MIL-88B(Fe) framework caused by washing with different solvents. Depending on the synthesis conditions, the obtained MIL-88B(Fe) crystals may exhibit different forms, ranging from hexagonal bipyramids (diamond-like crystals) to hexagonal prisms (rod-shaped crystals) [34].

The SEM images show that MIL-88B(Fe) has a hexagonal prismatic crystal structure (Figure 1b), with a length of approximately 300 nm and a width of 100 nm. It is distributed uniformly with smooth crystal surfaces. Due to the prolonged reaction time and temperature instability of the solvothermal synthesis method, smaller crystals of the same shape are also observed.

FTIR analysis shows that the characteristic peaks of MIL-88B(Fe) appear at 557 cm⁻^1^, 750 cm⁻^1^, 1394 cm⁻^1^, and 1600 cm⁻^1^ (Figure 1c). The peak at 557 cm⁻^1^ corresponds to Fe-O stretching vibrations in MIL-88B(Fe), while the absorption peaks at 1394 cm⁻^1^ and 1600 cm⁻^1^ are associated with symmetric and asymmetric stretching vibrations of dicarboxylate groups (organic ligands) in terephthalic acid, which are typical peaks of the MIL family [35]. The broad peak between 3400 and 3500 cm⁻^1^ is due to O-H stretching vibrations of adsorbed water on the surface, and the strong peak at 750 cm⁻^1^ is related to C-H bending vibrations of the benzene ring in the organic ligand [36], confirming the successful synthesis of MIL-88B(Fe).

The TGA curve explains the thermal stability of the sample in air [37]. As shown in Figure 1d, the mass loss of MIL-88B(Fe) occurs in three distinct stages with increasing temperature. The first weight loss takes place from 30 °C to 100 °C, which is attributed to the evaporation of physically adsorbed water and some residual solvents from the surface and pores of MIL-88B(Fe). The second weight loss occurring from 100 °C to 300 °C is reasonably associated with the pyrolysis of functional groups on the prepared MOF [38]. An additional weight loss observed at 380 °C corresponds to the decomposition of terephthalic acid within the MIL-88B(Fe) framework. At 600 °C, MIL-88B(Fe) completely decomposes, with the residual mass of iron oxide being approximately 27%, indicating good thermal stability of the prepared MIL-88B(Fe) [39].

### 3.2. Evaluation of the Bleaching Effect of MIL-88B(Fe)

We investigated the effects of bleaching temperature, bleaching time, and MIL-88B(Fe) dosage on the bleaching efficiency, pigments, nutrients, and harmful substances in *Idesia polycarpa* Maxim. oil (Table 1). The results showed that bleaching temperature had the highest impact on the content of trace substances in the oil, followed by MIL-88B(Fe) dosage, while the bleaching time demonstrated the low impact. When the bleaching temperature increased from 80 °C to 100 °C, the bleaching efficiency of *Idesia polycarpa* Maxim. oil increased from 49.99% to 91.89%, and further temperature increases resulted in a stable bleaching efficiency. At this point, the contents of lutein, β-carotene, and chlorophyll in the oil also reached their lowest values of 4.68 mg/kg, 0.03 mg/kg, and 6.14 mg/kg, respectively. The squalene content reached a minimum of 52.69 mg/100 g at 100 °C. Squalene, a precursor of phytosterols, was also found to be most affected by temperature in the study [40].

Şakir et al. compared the bleaching process of the sunflower oils using microwave and industrial techniques [41]. The bleaching efficiencies of microwave-assisted clay and industrial bleaching approaches were 83.76% and 85.68%, respectively. Md et al. optimized the clay-based bleaching process of corn oil for edible applications using a response surface methodology [42]. All the bleaching conditions showed red reduction (70–95%), yellow reduction (30–75%), β-carotene reduction (0–25%), and peroxide value (2–10 meq O_2_/kg). Claudio et al. evaluated the effect of different types (acid-activated and natural) and amounts (2% and 4%) of earths at different temperatures (60 °C and 80 °C) on bleaching hempseed oil [43]. The acid earths used alone or in combination with ultrasounds have shown a high efficiency in pigment removal in hemp oil, with the level of Vit. A reduced by 78–80%. For linseed oil, the ultrasound-assisted earth treatment led to a 50% reduction of Vit. Compared to the control groups. Our study showed that using the MIL-88B(Fe) achieved up to 91.89% bleaching efficiency of *Idesia polycarpa* Maxim. oil, and further temperature increases resulted in a stable bleaching efficiency. This indicates that the MIL-88B(Fe) material holds great promise for industrial applications.

The main sterols in *Idesia polycarpa* Maxim. oil are stigmasterol, β-sitosterol, and Δ5-avenasterol, with β-sitosterol being the most abundant at 249.21 mg/100 g in unbleached oil, which is higher than that in sunflower oil (57.58 mg/100 g), corn oil (151.65 mg/100 g), and tea seed oil (18.74 mg/100 g) [40]. As the bleaching temperature increased, the content of stigmasterol ranged from 13.56 mg/100 g to 14.67 mg/100 g, β-sitosterol from 211.89 mg/100 g to 219.35 mg/100 g, and Δ5-avenasterol from 54.85 mg/100 g to 58.68 mg/100 g, indicating a minimal effect of bleaching on the sterol content in the oil. The differences in sterol structure lie in the presence of double bonds and ethyl groups in the side chain at C17 [44]. The main difference between stigmasterol and β-sitosterol is the presence of double bonds, specifically, stigmasterol has a double bond at C5. The stigmasterol and Δ5-avenasterol are two position isomers. The high temperatures may cause the migration of the double bond between these molecules [45]. Moreover, Δ5-avenasterol is more stable than other phytosterols, possibly due to its long alkyl side chain.

Changes in bleaching time and MIL-88B(Fe) dosage resulted in stable levels of lutein, β-carotene, chlorophyll, squalene, and phytosterols. However, the FFA content was significantly influenced by MIL-88B(Fe) dosage. The bleached Idesia polycarpa Maxim. oil showed higher FFA levels, possibly because of MIL-88B(Fe)’s adsorbent properties [46]. The bleaching process showed that during the entire bleaching process, MIL-88B(Fe) effectively adsorbed carotenoids and chlorophyll from *Idesia polycarpa* Maxim. oil, changing the oil color from deep red to light yellow while maintaining the oil’s nutritional components. The content of the four types of FFAs increased after bleaching, which may be due to the high reaction temperature and the presence of unsaturated metal sites in the MIL-88B(Fe)’s structure, leading to the generation of more FFAs. Moreover, the high temperature can change the surface functional groups and increase the surface area of MOF, resulting in the high porosity. At high temperatures, the pigment molecules may penetrate the MOF material network more easily, increasing the adsorption capacity [47].

### 3.3. Adsorption Kinetics of MIL-88B(Fe)

Adsorption kinetics are used to reflect the evolution of the adsorption process over time. We measured the changes in the adsorption of MIL-88B(Fe) for carotenoids and chlorophyll within 0 to 85 min, obtaining the fitted curves of pseudo-first-order, pseudo-second-order, and intraparticle diffusion models, as shown in Figure 2. In the pseudo-first-order and pseudo-second-order curves, the adsorption capacity ($q_{t}$) of carotenoids (Figure 2a) rapidly increased within the first 35 min, reaching 70% of the maximum adsorption capacity. With a further extension of bleaching time, the adsorption rate of carotenoids gradually decreased and stabilized after 55 min. In contrast, the adsorption capacity ($q_{t}$) of chlorophyll (Figure 2b) increased more slowly compared to carotenoids, reaching 50% of the maximum adsorption capacity at 65 min. This suggests that carotenoids achieve equilibrium on the MIL-88B(Fe) surface more quickly, indicating stronger interactions between the adsorbent and carotenoids.

According to the kinetic equation fitting parameters (see Appendix A), the $R^{2}$ values indicate that the adsorption of carotenoids and chlorophyll on MIL-88B(Fe) is better described by the pseudo-first-order adsorption model. The calculated $q_{e}$ value for carotenoids (14.16 mg/g) aligns more closely with the experimental data (13.61 mg/g). Furthermore, the reaction rate constant of carotenoids (0.0486) is higher than that of chlorophyll (0.0079), suggesting that MIL-88B(Fe) has a stronger adsorption affinity for carotenoids compared to chlorophyll.

Weber and Morris proposed the intraparticle diffusion kinetic model to examine the rate-controlling steps in the adsorption [48]. The intraparticle diffusion model describes the adsorption process where the adsorption rate depends on the rate at which adsorbate molecules diffuse into the internal structure of the adsorbent particles (i.e., the process is diffusion-controlled) [49]. The intraparticle diffusion fitting plot for carotenoids (Figure 2c) shows three linear segments, while that for chlorophyll (Figure 2d) shows two linear segments. Generally, the number of linear segments in an adsorption plot is associated with the rate-controlling steps based on the pore structures and surface active sites of the adsorbents, which include bulk diffusion, liquid phase diffusion, intraparticle diffusion and physical or chemical reactions. Three linear segments suggest that the adsorption process is influenced by multiple rate-controlling steps, possibly including liquid phase diffusion, intraparticle diffusion, and chemical reactions. Two linear segments suggest that the adsorption process is mainly controlled by rate-limiting steps such as liquid phase diffusion and surface reactions, demonstrating that intraparticle diffusion is not the primary rate-controlling factor [50], and other factors such as surface reactions might play a more significant role. In fact, bulk diffusion and surface reactions are considered the fastest adsorption processes. The latter is determined by the intraparticle diffusion process.

This difference could arise from the distinct characteristics and reaction mechanisms of carotenoids and chlorophyll during the adsorption process. The coefficients $R^{2}$ for the three-stage and two-stage models gradually decreased (Appendix A) due to the strong dependence of intraparticle diffusion on solid-phase concentration. The adsorption process on MIL-88B(Fe) involves three independent steps: (1) Under a strong driving force, carotenoids and chlorophyll rapidly transfer from the solution to the MIL-88B(Fe) surface, occupying surface active sites; (2) Pigment molecules adsorbed on the MIL-88B(Fe) surface diffuse through the pores of MIL-88B(Fe) into the internal structure of the adsorbent; (3) As more pigment molecules penetrate MIL-88B(Fe), the diffusion process becomes increasingly hindered, and adsorption reaches equilibrium.

### 3.4. Adsorption Isotherms for MIL-88B(Fe)

Modeling experimental data through adsorption isotherm models is one of the most commonly used methods to study adsorption mechanisms. Isotherms describe the relationship between the equilibrium concentration of the adsorbate in the liquid phase and the equilibrium adsorption capacity in the solid phase at a specific temperature [51]. Figure 3 shows the fitting results of the Langmuir, Freundlich, Temkin, and Toth isotherm models, and the fitting parameters of the adsorption isotherms calculated from the fitted curves that are presented in Appendix A. The quality of fit between isotherm models and experimental data is generally evaluated based on the coefficient of determination ($R^{2}$), which represents the variance around the mean value. The closer the $R^{2}$ value is to 1, the better the model’s fitting performance.

In the fitting model for carotenoids, the Freundlich model had the highest $R^{2}$ value of 0.9766, while in the chlorophyll fitting model, the Temkin model had the highest $R^{2}$ value of 0.9480. The Freundlich and Temkin isotherm models well described the adsorption of carotenoids and chlorophyll, respectively, indicating multilayer and heterogeneous distribution at the adsorption sites [52]. The Freundlich model suggests that adsorption occurs first at the most energetic sites and then at the less energetic sites. The n coefficient of carotenoid with values greater than 1 indicated a good affinity between the pigment and the reactive MOF surface (Appendix A). The relatively high *R*^2^ value of the correlation coefficient (0.9766) for the Freundlich model also indicates that the model is suitable for describing the adsorption of carotenoids on the MIL-88B(Fe). These results imply an inhomogeneous surface or at least an inhomogeneous distribution of adsorption sites, and MIL-88B(Fe) is applicable to adsorb pigments in *Idesia polycarpa* Maxim. oil.

In the Langmuir model, the $R^{2}$ values for carotenoids and chlorophyll were 0.9604 and 0.7655, respectively, with maximum adsorption capacities of 1012 mg/g and 0.0221 mg/g, which showed significant deviation from the experimental data. These indicates that the adsorption of carotenoids and chlorophyll is not monolayer, and the adsorption surface of MIL-88B(Fe) is not homogeneous. The parameter n in the Freundlich equation is a measure of the linear deviation of adsorption, representing the degree of nonlinearity between concentration in solution and adsorption: when n = 1, the adsorption is linear; when n < 1, it is chemical adsorption; when n > 1, it indicates favorable physical adsorption [53]. In the Freundlich model, the $K_{F}$ value for carotenoids was 0.2151, and the n value was 1.128, demonstrating that the MIL-88B(Fe) adsorption of carotenoids is favorable physical adsorption. In the Temkin model, the $K_{T}$ value for chlorophyll was -0.0045 g/kg, and the $B_{T}$ value was 0.0518 kJ/mol, suggesting that the MIL-88B(Fe) adsorption of chlorophyll is physical adsorption. When the bond energy value is less than 8 kJ/mol, the involved adsorption mechanism is physical adsorption [54]. The t parameter in the Toth model reflects the temperature dependence of the isotherm parameter $Q_{T}$, indicating the degree of system heterogeneity. The t values for carotenoids and chlorophyll were 58.04 and −1.060, respectively, primarily due to the intensity of the interactions between the adsorbate and adsorbent in chemical adsorption.

### 3.5. Adsorption Thermodynamics of MIL-88B(Fe)

We also investigated the impact of temperature on the adsorption of MIL-88B(Fe) in the range of 80 °C to 120 °C. The values of Gibbs free energy change (ΔG^0^) and entropy change (ΔS^0^) for the adsorption of carotenoids and chlorophyll on MIL-88B(Fe) were measured based on the slope and intercept of the plot of ln$k_{d}$ versus 1/T (not shown in the figure), and the thermodynamic parameters were calculated. According to Appendix A, the ΔG^0^ values for carotenoids calculated in the temperature range of 353 to 393 K were −19.87 kJ mol^−1^ to −12.14 kJ mol^−1^, while the ΔG^0^ values for chlorophyll ranged from −10.50 kJ mol^−1^ to −4.994 kJ mol^−1^, with ΔG^0^ gradually decreasing as the temperature increased. The Gibbs free energy (ΔG^0^) is an indicator of the spontaneity of a chemical reaction and is an important criterion for spontaneity [55]. Both carotenoids and chlorophyll had negative ΔG^0^ values, indicating the feasibility and spontaneity of the adsorption process. The ΔG^0^ values for carotenoids were lower than those for chlorophyll, implying that the MIL-88B(Fe) adsorption of carotenoids is a more spontaneous process. The enthalpy change (ΔH^0^) values for carotenoids and chlorophyll were 64.00 kJ mol^−1^ and 45.34 kJ mol^−1^, respectively, indicating the endothermic nature of the adsorption process. Enthalpy change (ΔH^0^) can be used to describe thermodynamic changes in processes such as chemical reactions and phase transitions [56]. The entropy change (ΔS^0^) values for carotenoids and chlorophyll were both positive, at 0.1578 J mol^−1^ K^−1^ and 0.0851 J mol^−1^ K^−1^, respectively, suggesting that the surface activity of MIL-88B(Fe) increased, leading to higher affinity for the pigments and increased randomness at the solid-liquid interface during the adsorption process [57].

It is recognized that temperature has a significant influence on adsorption processes and capacity, which is associated with the kinetic energy, solubility, mobility, and chemical potential of the adsorbate [58]. In general, higher temperatures increase the adsorption capacity of an adsorbent. The increase in the adsorption capacity of MOF with increasing temperature may be due to the fact that higher temperatures enhance the fluidity of the solution and its diffusion into the adsorbent material, or increase the surface area of the adsorbent material, resulting in the creation of more active adsorption sites. However, high temperatures of more than 100 °C may be difficult to meet the criteria of availability, and environmental and economic feasibility.

### 3.6. Reusability and Safety of MIL-88B(Fe)

To evaluate the potential of MIL-88B(Fe) in practical applications, it is necessary to study its stability and reusability, as stable adsorbents can be recycled to save costs [59]. In this study, Soxhlet extraction was initially used to wash and regenerate the MIL-88B(Fe) after adsorption. The bleaching rate of MIL-88B(Fe) decreased to 58.89% after three cycles, indicating a significant reduction in bleaching efficiency. During the washing process, some *Idesia polycarpa* Maxim. oil adsorbed on MIL-88B(Fe) was difficult to remove completely, leading to a low recovery rate of MIL-88B(Fe). Therefore, the adsorbed MIL-88B(Fe) was washed under magnetic stirring with ethanol at 60 °C. Table 2 showed that the bleaching rate of MIL-88B(Fe) remained at 97.37% after four cycles, demonstrating good adsorption stability and reusability. However, the bleaching rate dropped to 33.83% in the fifth cycle and to 3.87% in the sixth cycle, possibly because MIL-88B(Fe) approached or reached adsorption saturation after multiple cycles and could not be fully regenerated, resulting in a significant decline in adsorption capacity. Additionally, impurities accumulated on the MIL-88B(Fe)’s surface during the adsorption process, causing blockages that affected its performance. Previous studies have shown that MIL-88B(Fe) exhibits good reusability when adsorbing dyes in water [60]. Therefore, due to its low cost, simple synthesis, and good reusability, MIL-88B(Fe) significantly reduces the cost of bleaching *Idesia polycarpa* Maxim. oil, making it advantageous for applications in the oil refining industry. The extracted β-carotenes can be potentially used as soluble orange-red pigments in the food industry, while also can be utilized in the realms of pharmaceuticals and cosmetics [61].

The determination of Fe content in the bleached *Idesia polycarpa* Maxim. oil showed that no Fe ion was detected in each digestion solution/sample solution, indicating that MIL-88B(Fe) maintained structural stability during the bleaching process. It was completely separated after bleaching and did not remain in the oil. The composition and biodegradability of the MIL-88B(Fe) structure are advantageous in terms of toxicology because it contains iron as the metal source, which is far less toxic compared to other metals such as cobalt [62]. To date, MOF materials have significant value in the food sector and have been developed for applications in removing food contaminants, food packaging, food storage, and food separation and purification due to their multifunctionality [63].

## 4. Conclusions

In this study: MIL-88B(Fe) was synthesized and used as an adsorbent for the bleaching of *Idesia polycarpa* Maxim. oil. It showed a porous adsorbent material with a length of approximately 300 nm and width of 100 nm and exhibited good thermal stability. The effects of bleaching on the pigments, nutrients, and harmful substances in *Idesia polycarpa* Maxim. oil were investigated to explore the adsorption selectivity of MIL-88B(Fe). When the bleaching temperature increased from 80 °C to 100 °C, the bleaching efficiency of *Idesia polycarpa* Maxim. oil increased from 49.99% to 91.89%. The content of stigmasterol ranged from 13.56 mg/100 g to 14.67 mg/100 g, β-sitosterol from 211.89 mg/100 g to 219.35 mg/100 g, and Δ5-avenasterol from 54.85 mg/100 g to 58.68 mg/100 g, suggesting the bleaching process had little effect on the nutrients in the oil. The adsorption process was modeled and analyzed, based on the kinetics, isotherms, and thermodynamics of carotenoid and chlorophyll. The results showed that MIL-88B(Fe) exhibited good adsorption selectivity for carotenoids, particularly lutein, in *Idesia polycarpa* Maxim. oil. The content of lutein decreased from 118.27 mg/kg to 0.01 mg/kg after 65 min of bleaching, while squalene and phytosterols in the oil were well preserved. The content of free fatty acids slightly increased due to the influence of open iron metal sites in the MIL-88B(Fe) structure as well as the temperature used. Moreover, the adsorption of carotenoids and chlorophylls could be described by the pseudo-first-order kinetic model in which the MIL-88B(Fe) showed a stronger affinity for carotenoids compared to chlorophylls. This was based on the calculated *q*_e_ value for carotenoids (14.16 mg/g) more closely with the experimental data (13.61 mg/g). The reaction rate constant of carotenoids (0.0486) was also higher than that of chlorophyll (0.0079). The intraparticle diffusion plots for carotenoids and chlorophylls exhibited three or two linear segments, indicating that intraparticle diffusion was not the sole rate-limiting mechanism and that the adsorption process involved multiple independent steps. Isotherm analysis revealed that the Freundlich model better described carotenoid adsorption, suggesting multilayer adsorption with heterogeneous distribution; however, the Temkin model was more suitable for describing chlorophyll adsorption, indicating multilayer physical adsorption with linearly decreasing adsorption heat among the molecules in the layer. Thermodynamic studies indicated that the adsorption of carotenoids and chlorophylls on MIL-88B(Fe) was spontaneous and endothermic. Overall, this study demonstrates that MIL-88B(Fe) can effectively adsorb carotenoids and chlorophylls in the *Idesia polycarpa* Maxim. oil and exhibit good reusability and safety. It holds great potential for potential large-scale implementation of bleaching for vegetable oils as a practical and cost-effective process. Future research should focus on the effect of high temperatures on the stability of MIL-88B(Fe) materials and how to improve the adsorption efficiency of MOF for pigments in edible oils under low-temperature conditions.

## Figures and Tables

**Figure 1 foods-14-00787-f001:**
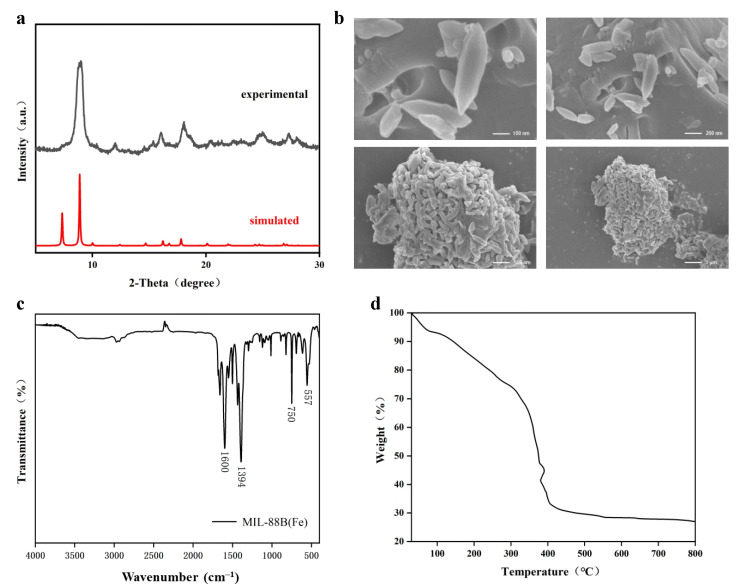
Characterization of MIL-88B(Fe). XRD (**a**), SEM (**b**), FT-IR (**c**), and TGA (**d**) patterns of the prepared MIL-88B(Fe). In SEM (**b**), the synthesized MIL-88B(Fe) was observed at the scales of 100 nm, 200 nm, 500 nm and 1 μm, respectively.

**Figure 2 foods-14-00787-f002:**
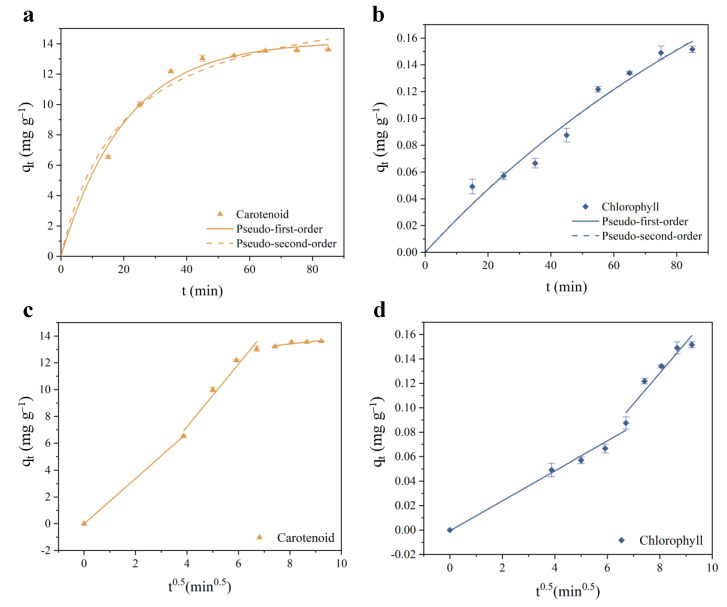
Modeling of carotenoid and chlorophyll adsorption processes using pseudo-first-order, pseudo-second-order, and intraparticle diffusion kinetic models. Adsorption kinetic curves of carotenoid (**a**) and chlorophyll (**b**) are plotted. Intraparticle diffusion plots of carotenoid (**c**) and chlorophyll (**d**) are shown.

**Figure 3 foods-14-00787-f003:**
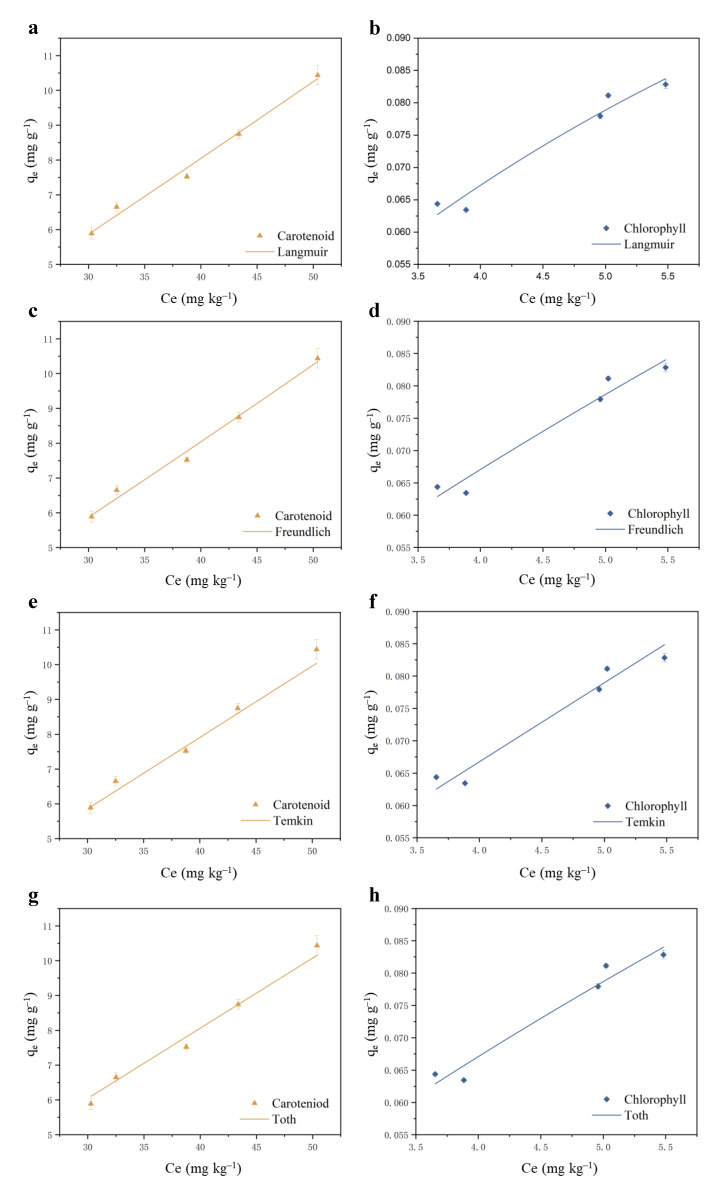
Modeling of carotenoid and chlorophyll adsorption processes using Langmuir (**a**,**b**), Freundlich (**c**,**d**), Temkin (**e**,**f**) and Toth (**g**,**h**) isotherms.

**Table 1 foods-14-00787-t001:** Effect of MIL-88B(Fe) on bleaching efficiency, pigment content, squalene, phytosterol and free fatty acid content of *Idesia polycarpa* Maxim. oil.

Bleaching Time (min)	Bleaching Temperature (℃)	MIL-88B(Fe) Addition (*w*/*v*%)	Bleaching Efficiency (%)	Lutein (mg/kg)	β-Carotene (mg/kg)	Chlorophyll (mg/kg)	Squalene (mg/100g)	Phytosterol (mg/100g)			FFA (mg/100g)			
								Stigmasterol	β-Sitosterol	Δ5-Avenasterol	C18:0	C18:1	C18:2	C18:3
Unbleached oil	/			118.27 ± 1.38 ^a^	1.32 ± 0.05 ^a^	9.92 ± 0.17 ^a^	58.44 ± 1.65 ^abc^	15.66 ± 0.12 ^bc^	249.21 ± 0.80 ^a^	60.85 ± 9.35 ^a^	0.08 ± 0.02 ^h^	0.39 ± 0.05 ^cd^	0.10 ± 0.03 ^h^	0.09 ± 0.00 ^cde^
35	100	5	88.81 ± 0.49 ^de^	5.76 ± 1.46 ^efg^	0.04 ± 0.01 ^ef^	6.57 ± 0.20 ^cd^	53.63± 0.69 ^efg^	21.63 ± 4.79 ^a^	246.58 ± 11.86 ^ab^	57.05 ± 13.50 ^a^	0.19 ± 0.03 ^cd^	0.60 ± 0.04 ^b^	0.56 ± 0.02 ^b^	0.16 ± 0.02 ^a^
45			95.49 ± 1.99 ^abc^	2.35 ± 1.39 ^ghi^	0.02 ± 0.00 ^efg^	5.37 ± 0.31 ^efg^	55.16 ± 2.34 ^cdefg^	21.23 ± 4.53 ^a^	244.76 ± 10.57 ^abc^	52.93 ± 1.81 ^a^	0.33 ± 0.02 ^b^	0.58 ± 0.09 ^b^	0.45 ± 0.01 ^bcde^	0.19 ± 0.00 ^a^
55			95.43 ± 0.41 ^abc^	0.59 ± 0.01 ^hi^	0.02 ± 0.00 ^fg^	3.11 ± 0.26 ^ij^	61.16 ± 0.81 ^a^	14.82 ± 0.20 ^bc^	236.28 ± 2.80 ^abcd^	51.34 ± 2.70 ^a^	0.43 ± 0.02 ^a^	0.55 ± 0.09 ^b^	0.51 ± 0.04 ^bcd^	0.17 ± 0.05 ^a^
65			97.67 ± 0.56 ^a^	0.01 ± 0.01 ^i^	n.d.	1.47 ± 0.58 ^k^	56.56 ± 4.69 ^abcdef^	14.83 ± 2.92 ^bc^	228.46 ± 7.29 ^cdef^	55.36 ± 1.64 ^a^	0.11 ± 0.03 ^fgh^	0.40 ± 0.08 ^cd^	0.40 ± 0.04 ^bcdef^	0.06 ± 0.00 ^de^
75			96.59 ± 0.43 ^ab^	0.07 ± 0.03 ^i^	n.d.	2.90 ± 0.62 ^j^	59.16 ± 4.30 ^abcd^	15.80 ± 0.77 ^bc^	235.22 ± 1.60 ^abcd^	54.24 ± 6.81 ^a^	0.11 ± 0.02 ^efgh^	0.32 ± 0.04 ^d^	0.36 ± 0.02 ^defg^	0.08 ± 0.00 ^cde^
35	80	5	49.99 ± 0.01 ^g^	91.26 ± 0.74 ^b^	0.80 ± 0.01 ^b^	7.65 ± 0.13 ^b^	60.51 ± 1.89 ^ab^	14.62 ± 0.31 ^ab^	219.35 ± 2.53 ^abcde^	58.68 ± 4.42 ^c^	0.20 ± 0.03 ^c^	0.48 ± 0.09 ^bc^	0.28 ± 0.06 ^fg^	0.17 ± 0.01 ^a^
	90		59.12 ± 2.61 ^f^	52.09 ± 3.05 ^c^	0.41 ± 0.02 ^c^	7.34 ± 0.09 ^bc^	59.34 ± 1.00 ^abc^	14.67 ± 0.81 ^bc^	212.01 ± 2.68 ^f^	54.85 ± 3.32 ^c^	0.13 ± 0.02 ^efgh^	0.48 ± 0.05 ^bc^	0.30 ± 0.08 ^efg^	0.15 ± 0.01 ^ab^
	100		91.89 ± 1.14 ^cd^	4.68 ± 1.85 ^fg^	0.03 ± 0.01 ^efg^	6.14 ± 0.40 ^de^	52.69 ± 1.81 ^fg^	13.56 ± 0.93 ^bc^	211.89 ± 2.02 ^f^	56.00 ± 3.69 ^c^	0.09 ± 0.00 ^gh^	0.48 ± 0.00 ^bc^	0.23 ± 0.06 ^gh^	0.12 ± 0.03 ^bc^
	110		86.66 ± 5.18 ^e^	13.16 ± 3.14 ^d^	0.09 ± 0.03 ^d^	7.30 ± 0.13 ^bc^	54.46 ± 0.69 ^defg^	14.59 ± 0.83 ^bc^	216.67 ± 7.94 ^ef^	58.06 ± 6.81 ^bc^	0.11 ± 0.00 ^fgh^	0.46 ± 0.01 ^bc^	0.30 ± 0.03 ^efg^	0.09 ± 0.00 ^cde^
	120		91.59 ± 1.20 ^cd^	6.09 ± 0.38 ^ef^	0.04 ± 0.00 ^ef^	7.33 ± 0.01 ^bc^	50.97 ± 0.99 ^g^	14.18 ± 0.41 ^bc^	216.26 ± 3.87 ^ef^	57.23 ± 4.82 ^c^	0.14 ± 0.01 ^defg^	0.48 ± 0.02 ^a^	0.27 ± 0.01 ^fg^	0.10 ± 0.02 ^cd^
35	100	5	92.21 ± 2.48 ^bcd^	8.96 ± 0.90 ^e^	0.06 ± 0.01 ^e^	5.49 ± 0.02 ^ef^	55.94 ± 1.16 ^bcdef^	13.58 ± 2.18 ^bc^	231.21 ± 3.07 ^bcde^	57.03 ± 2.32 ^a^	0.17 ± 0.02 ^cde^	0.60 ± 0.03 ^b^	0.47 ± 0.04 ^bcd^	0.08 ± 0.08 ^cde^
		6	92.68 ± 1.21 ^bcd^	4.79 ± 1.72 ^fg^	0.04 ± 0.00 ^ef^	4.77 ± 0.27 ^fgh^	55.51 ± 1.29 ^cdefg^	14.00 ± 3.45 ^bc^	240.28 ± 1.20 ^abc^	55.02 ± 0.50 ^a^	0.19 ± 0.04 ^cd^	0.84 ± 0.10 ^a^	0.75 ± 0.16 ^a^	0.09 ± 0.02 ^cde^
		7	93.79 ± 0.43 ^abc^	3.80 ± 0.91 ^fgh^	0.02 ± 0.00 ^efg^	5.12 ± 0.14 ^fg^	54.34 ± 0.79 ^defg^	11.49 ± 0.57 ^c^	221.84 ± 1.80 ^def^	55.49 ± 5.36 ^a^	0.14 ± 0.01 ^efg^	0.51 ± 0.00 ^bc^	0.39 ± 0.04 ^cdef^	0.08 ± 0.02 ^cde^
		8	93.10 ± 0.71 ^bcd^	3.86 ± 0.58 ^fgh^	0.03 ± 0.01 ^efg^	4.49 ± 1.17 ^gh^	57.88 ± 1.09 ^abcde^	12.51 ± 1.01 ^bc^	219.62 ± 3.82 ^def^	49.83 ± 1.40 ^ab^	0.16 ± 0.02 ^cdef^	0.57 ± 0.05 ^b^	0.45 ± 0.12 ^bcde^	0.07 ± 0.00 ^de^
		9	93.70 ± 0.56 ^abc^	3.36 ± 2.02 ^fghi^	0.03 ± 0.00 ^efg^	3.93 ± 0.06 ^hi^	56.95 ± 1.70 ^abcdef^	15.33 ± 0.46 ^bc^	228.76 ± 9.47 ^cdef^	56.99 ± 2.15 ^a^	0.14 ± 0.01 ^defg^	0.59 ± 0.05 ^b^	0.53 ± 0.10 ^bc^	0.05 ± 0.00 ^e^

Note: different letters indicate statistically significant differences (*p* < 0.05).

**Table 2 foods-14-00787-t002:** Reusability of MIL-88B(Fe).

Washing Method	First Use	Second Use	Third Use	Fourth Use	Fifth Use	Sixth Use
Soxhlet extraction	96.46 ± 0.00 ^a^	96.21 ± 0.07 ^a^	58.89 ± 0.12 ^b^	/		
60 °C ethanol	96.46 ± 0.00 ^a^	97.55 ± 0.43 ^a^	96.90 ± 2.45 ^a^	97.37 ± 0.23 ^a^	33.83 ± 0.16 ^b^	3.87 ± 1.34 ^c^

Note: different letters indicate statistically significant differences (*p* < 0.05).

## Data Availability

Data will be made available on request.

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
