# Peer review of "Bleaching of *Idesia polycarpa* Maxim. Oil Using a Metal-Organic Framework-Based Adsorbent: Kinetics and Adsorption Isotherms"

_foods, 2025, doi:10.3390/foods14050787_

Round 1
Reviewer 1 Report
Comments and Suggestions for Authors
Line 27: I believe introduction would benefit from more recent references.
Line 76: Does other MOF have the same feature? Was this the reason to select this MOF? Please, provide a more detailed rationale for selecting MIL-88B(Fe) over other MOFs or adsorbents.
Line 117: Please, clarify the range of adsorbent dosages and temperatures selected for the bleaching experiments. Is the amount related to the pigment content?
Line 121: what method did you use for the separation?
Line 326: Please provide how the bleaching efficiency of MIL-88B(Fe) compares to other commercial bleaching methods.
Line 354: Please provide macromolecular insights into how the processing parameters affect the bleaching.
Line 376: considering the assumptions of each model, what does it mean that the pseudo-first order fits the data best?
Line 390: Please, provide more details regarding the implications of rate-controlling steps (e.g., surface reactions vs. intraparticle diffusion
Line 427: Please, clarify considering the assumptions of each model, what are the implications Freundlich and Temkin models to best describe the adsorption behavior. Include implications for adsorbent design.
Line 455: Please, discuss how temperature influences adsorption feasibility.
Line 490: can the extracted pigments namely beta-carotens be further used for other applications ?
Line 492: Was the Fe content determined after the first extraction? Do you believe this Fe stability is extendable for the remaining extractions?
Table 1 is hard to read, I suggest dividing this table into 2
Figure 2 and Figure 3 are really hard to read
Table 2 should be around line 491, since line 416 calls for supplementary table 2 and not table 2.
I think the formatting of the reference is wrong since its not numbered.
Author Response
Line 27: I believe introduction would benefit from more recent references.
Response: More recent references have been added to the introduction section of the revised manuscript.
Line 76: Does other MOF have the same feature? Was this the reason to select this MOF? Please, provide a more detailed rationale for selecting MIL-88B(Fe) over other MOFs or adsorbents.
Response: We have provided more reasons for selecting the MIL-88B(Fe) in the introduction section of the revised manuscript (lines 82-86).
Line 117: Please, clarify the range of adsorbent dosages and temperatures selected for the bleaching experiments. Is the amount related to the pigment content?
Response: We have clarified the ranges of adsorbent dosages and temperatures used for bleaching in the revised manuscript (lines 147-149). The adsorbent amount is based on our preliminary tests and some previous studies (e.g., Monique et al.)
Monique et al. Bleaching optimization and winterization step evaluation in the refinement of rice bran oil. Separation and Purification Technology, 175,2017, 72-78.
Line 121: what method did you use for the separation?
Response: We used vacuum filtration to separate the oil from the adsorbent, and we have provided more details in the revised manuscript (line 150).
Line 326: Please provide how the bleaching efficiency of MIL-88B(Fe) compares to other commercial bleaching methods.
Response: We have compared the bleaching efficiency of MIL-88B(Fe) with some previously reported bleaching methods in the revised manuscript (lines 368-382).
Line 354: Please provide macromolecular insights into how the processing parameters affect the bleaching.
Response: We have provided more discussions on how the processing parameters affect the bleaching in the revised manuscript (lines 439-442).
Line 376: considering the assumptions of each model, what does it mean that the pseudo-first order fits the data best?
Response: Fig. 2a and 2b show the fitted curves of pseudo-first-order and pseudo-second-order models. Based on the supplementary table 1, the R2 values of the pseudo-first-order are higher than that of the pseudo-second-order model, indicating the adsorption curve fits better pseudo-first-order kinetic model (line 456).
Line 390: Please, provide more details regarding the implications of rate-controlling steps (e.g., surface reactions vs. intraparticle diffusion
Response: More details regarding the implications of rate-controlling steps have been provided in the revised manuscript (lines 476-485).
Line 427: Please, clarify considering the assumptions of each model, what are the implications Freundlich and Temkin models to best describe the adsorption behavior. Include implications for adsorbent design.
Response: We have clarified more implications of models for adsorbent selection and application in the revised manuscript (lines 515-523).
Line 455: Please, discuss how temperature influences adsorption feasibility.
Response: We have discussed the influences of processing temperature on adsorption feasibility in the manuscript (lines 564-572).
Line 490: can the extracted pigments namely beta-carotens be further used for other applications ?
Response: We have added more potential applications of the extracted pigments in the revised manuscript (lines 592-594).
Line 492: Was the Fe content determined after the first extraction? Do you believe this Fe stability is extendable for the remaining extractions?
Response: The Fe content was determined for each extraction, thus the MIL-88B(Fe) was believed to maintain structural stability during the bleaching process, without releasing Fe in the oil.
Table 1 is hard to read, I suggest dividing this table into 2
Response: We have updated Table 1 to make it look clearer in the revised manuscript (line 413).
Figure 2 and Figure 3 are really hard to read
Response: We have re-drawn Figures 2 and 3 in the revised manuscript (line 464 and 508).
Table 2 should be around line 491, since line 416 calls for supplementary table 2 and not table 2.
Response: Table 2 has been moved to the right position around line 566 in the revised manuscript (line 595).
I think the formatting of the reference is wrong since its not numbered.
Response: The reference formatting has been corrected according to the MDPI requirement.
Reviewer 2 Report
Comments and Suggestions for Authors The manuscript addresses an important and novel topic: the use of MOFs. The following are observations to improve the article.
1. The title can be improved by emphasizing the study's novelty, making it more appealing to readers.
2. The abstract should include what the abbreviation MOF stands for (line 32) and other abbreviations used. It should also include the impact of the research results on the food industry.
3. The keywords (line 49) should differ from those in the title.
4. In the first paragraph of the introduction (lines 58-64), it is recommended to write the central problem and emphasize the study's novelty to capture the reader's attention. In the following paragraphs, the hypotheses and objectives of the study should be included in an orderly manner to organize the article better and make the reading flow smoothly.
5. In section 2.1 of the materials (lines 122-133), more details of the materials should be provided, such as the oil. In the case of the reagents, include the brand, city, and country in parentheses. Also, check the same with the equipment and instruments omitted in the materials and methods section.
6. Section 2.2 (lines 144-150) should include the methodology used for the X-ray diffraction, SEM, TGA, and FTIR analyses. For example, include which equipment was used, the methods used in detail, and the sources used in the methodologies.
7. In section 2.3, bleaching experiments (line 151), the type of experimental design used should be explicitly stated. This will improve the presentation of the results and make the statistical analysis more robust.
8. The equations used should be listed (lines 176, 185, 216-217, 229-231, 247-250, 276-280, 305-306). As the submitted format is not the template used in FOODs, it is recommended that you download the template and write the manuscript according to the journal's guidelines.
9. In section 3 of the results and discussions, it is recommended that, based on previous observations on the objectives and experimental design, the presentation of this section be better organized by performing a more critical analysis of current scientific articles. Please don't make the discussions only comparative; go deeper into the physical and chemical mechanisms involved.
10. It is necessary to improve the quality of Figure 1 because there is a lot of noise in the X-ray diffraction, FTIR, and TGA images. In the case of DX and TGA, detailed information about what happens in them should be included in the pictures. In the case of SEM, it is impossible to see in detail the conditions in which it was measured or the material's morphology.
11. Table 1 should be improved since the information presented is confusing and disordered.
12. To improve the presentation of the results, it is recommended that Figures 2 and 3 be colored.
13. The results and discussions are not integrated, which makes it difficult to follow a logical narrative between the experimental findings and their scientific meaning.
14. It is recommended that the key findings obtained in the research be summarized at the end of the results and discussions section.
15. The conclusions are too general and lack a direct connection to the objectives and key findings obtained in the research. It is also recommended that specific quantitative results be included, limitations discussed, and future research related to practical applications proposed.
16. Decrease the ithenticate similarity index (35%).
Author Response
The manuscript addresses an important and novel topic: the use of MOFs. The following are observations to improve the article.
1.The title can be improved by emphasizing the study's novelty, making it more appealing to readers.
Response: We have updated the title according to the reviewer’s suggestions in the revised manuscript (line 1). The new title is “Bleaching of Idesia polycarpa Maxim. Oil Using a Metal-Organic Framework-Based Adsorbent: Kinetics and Adsorption Isotherms”.
- The abstract should include what the abbreviation MOF stands for (line 32) and other abbreviations used. It should also include the impact of the research results on the food industry.
Response: We have provided the abbreviation of MOF and highlighted the impact of our MOF materials on the vegetable oil industry (lines 12, 25).
- The keywords (line 49) should differ from those in the title.
Response: We have updated the keywords in the revised manuscript (line 27).
- In the first paragraph of the introduction (lines 58-64), it is recommended to write the central problem and emphasize the study's novelty to capture the reader's attention. In the following paragraphs, the hypotheses and objectives of the study should be included in an orderly manner to organize the article better and make the reading flow smoothly.
Response: We have reorganized the introduction section to make it read smoothly in the revised manuscript.
- In section 2.1 of the materials (lines 122-133), more details of the materials should be provided, such as the oil. In the case of the reagents, include the brand, city, and country in parentheses. Also, check the same with the equipment and instruments omitted in the materials and methods section.
Response: We have provided more details of the materials used in this study in the material section of the revised manuscript.
- Section 2.2 (lines 144-150) should include the methodology used for the X-ray diffraction, SEM, TGA, and FTIR analyses. For example, include which equipment was used, the methods used in detail, and the sources used in the methodologies.
Response: More details of the methodology for XRD, SEM, TGA, and FTIR have been provided in the section “2.2. Synthesis and Characterization of MOF Adsorbents” of the revised manuscript (lines 127-145).
- In section 2.3, bleaching experiments (line 151), the type of experimental design used should be explicitly stated. This will improve the presentation of the results and make the statistical analysis more robust.
Response: We have provided more details on the experimental design in the revised manuscript (lines 147-152).
- The equations used should be listed (lines 176, 185, 216-217, 229-231, 247-250, 276-280, 305-306). As the submitted format is not the template used in FOODs, it is recommended that you download the template and write the manuscript according to the journal's guidelines.
Response: We have made all of these equations listed in the revised manuscript.
- In section 3 of the results and discussions, it is recommended that, based on previous observations on the objectives and experimental design, the presentation of this section be better organized by performing a more critical analysis of current scientific articles. Please don't make the discussions only comparative; go deeper into the physical and chemical mechanisms involved.
Response: We have provided deeper discussions into the oil bleaching by MIL-88B(Fe) in the revised manuscript (lines 391-394, 439-442, 515-523, 564-572).
- It is necessary to improve the quality of Figure 1 because there is a lot of noise in the X-ray diffraction, FTIR, and TGA images. In the case of DX and TGA, detailed information about what happens in them should be included in the pictures. In the case of SEM, it is impossible to see in detail the conditions in which it was measured or the material's morphology.
Response: We have improved the quality of Figure 1 in the revised manuscript. A detailed description of these figures has been provided in the section “3.1. Synthesis and Characterization of MIL-88B(Fe)” in the revised manuscript.
- Table 1 should be improved since the information presented is confusing and disordered.
Response: We have updated Table 1 to make it clearer in the revised manuscript (line 413).
- To improve the presentation of the results, it is recommended that Figures 2 and 3 be colored.
Response: Figures 2 and 3 have been colored in the revised manuscript.
- The results and discussions are not integrated, which makes it difficult to follow a logical narrative between the experimental findings and their scientific meaning.
Response: We have integrated several paragraphs to make it logical in the revised manuscript.
- It is recommended that the key findings obtained in the research be summarized at the end of the results and discussions section.
Response: The key findings of this study are summarized in the conclusion section of the revised manuscript (lines 612-616, 625-628).
- The conclusions are too general and lack a direct connection to the objectives and key findings obtained in the research. It is also recommended that specific quantitative results be included, limitations discussed, and future research related to practical applications proposed.
Response: We have provided more key findings and followed several recommendations according to the reviewer’s comments in the conclusion section of the revised manuscript (lines 638-642).
- Decrease the ithenticate similarity index (35%).
Response: The similarity of this text has been decreased.
Reviewer 3 Report
Comments and Suggestions for Authors
1The abbreviation MOF was introduced in the Abstract - explain what it refers to. In addition, highlight the significant results concerning this study in this section
In Introduction part emphasize novelty and scientific contribution of this study.
Number the equations throughout the entire manuscript
Equations 3 and 4 are well known. it is not necessary to assign them a reference: (Igansi, Engelmann, Lütke, Porto, Pinto, & Cadaval Jr, 179 2019)
In the case of equations for kinetic models (5) – (7), the reference of the author who first derived these equations should be given. Citation of Gil, Amiri, Abedi-Koupai, & Eslamian, 2018 is incorrect.
Provide reference for the isothermal models used.
Provide reference for the equations for calculation of thermodynamics parameters.
Provide equation for reusability calculation
Figure 2. specify in the description of the figure what a, b, c, and d refers to
Figure 3. explain why two images are shown for each model
present graphically and tabularly the values of standard deviations (isotherm and kinetic study)
The authors should compare their own results with available literature. Only in this way is it possible to indicate the added value for the new material.
In the conclusion part highlight what the findings of this study contribute and where they can be applied. More scientific conclusion is needed.
Author Response
The abbreviation MOF was introduced in the Abstract - explain what it refers to. In addition, highlight the significant results concerning this study in this section
Response: The full name of MOF has been provided in the abstract. We have highlighted more significant results in this section according to the reviewer’s suggestions in the revised manuscript (lines 12, 18).
In Introduction part emphasize novelty and scientific contribution of this study.
Response: We have highlighted the novelty and scientific contribution of this work in the introduction section of the revised manuscript (lines 87-95).
Number the equations throughout the entire manuscript
Response: All the equations throughout the entire text have been numbered in the revised manuscript.
Equations 3 and 4 are well known. it is not necessary to assign them a reference: (Igansi, Engelmann, Lütke, Porto, Pinto, & Cadaval Jr, 179 2019)
Response: The reference has been removed in the revised text.
In the case of equations for kinetic models (5) – (7), the reference of the author who first derived these equations should be given. Citation of Gil, Amiri, Abedi-Koupai, & Eslamian, 2018 is incorrect.
Response: We have corrected the reference to these equations in the revised manuscript (line 220).
Provide reference for the isothermal models used.
Response: We have provided a reference for the isothermal models in the revised manuscript (line 239).
Provide reference for the equations for calculation of thermodynamics parameters.
Response: We have provided more references for the calculation of thermodynamics parameters in the main text (line 266).
Provide equation for reusability calculation
Response: The equation for reusability calculation has been added in the section “2.11. Determination of Reusability of MIL-88B(Fe)” of the revised manuscript (line 288).
Figure 2. specify in the description of the figure what a, b, c, and d refers to
Response: We have specified the description of Figure 2 a-d in the revised manuscript (lines 466-468).
Figure 3. explain why two images are shown for each model
Response: We have re-drawn Figure 3, and two images are used for carotenoid and chlorophyll adsorption processes, respectively, under each model.
present graphically and tabularly the values of standard deviations (isotherm and kinetic study)
Response: Standard deviations have been provided in figures 2 and 3 in the revised manuscript.
The authors should compare their own results with available literature. Only in this way is it possible to indicate the added value for the new material.
Response: We have made comparisons of MOF material used with other adsorbents such as clay, and earth in the section “3.2. Evaluation of the Bleaching Effect of MIL-88B(Fe)” of the revised manuscript.
In the conclusion part highlight what the findings of this study contribute and where they can be applied. More scientific conclusion is needed.
Response: We have provided more scientific conclusions in the revised manuscript.
Round 2
Reviewer 3 Report
Comments and Suggestions for Authors
The authors corrected the Manuscript according to reviewers' suggestions.